# Biomarkers for Early Detection, Prognosis, and Therapeutics of Esophageal Cancers

**DOI:** 10.3390/ijms24043316

**Published:** 2023-02-07

**Authors:** Vikrant Rai, Joe Abdo, Devendra K. Agrawal

**Affiliations:** 1Department of Translational Research, Western University of Health Sciences, 309 E. Second Street, Pomona, CA 91766, USA; 2Department of Research and Development, Stella Diagnostics, Inc., Salt Lake City, UT 84104, USA; 3ProPhase Labs, Inc., Garden City, NY 11530, USA

**Keywords:** esophageal carcinoma, esophageal adenocarcinoma, biomarkers, blood, urine, saliva

## Abstract

Esophageal cancer (EC) is the deadliest cancer worldwide, with a 92% annual mortality rate per incidence. Esophageal squamous cell carcinoma (ESCC) and esophageal adenocarcinoma (EAC) are the two major types of ECs, with EAC having one of the worst prognoses in oncology. Limited screening techniques and a lack of molecular analysis of diseased tissues have led to late-stage presentation and very low survival durations. The five-year survival rate of EC is less than 20%. Thus, early diagnosis of EC may prolong survival and improve clinical outcomes. Cellular and molecular biomarkers are used for diagnosis. At present, esophageal biopsy during upper endoscopy and histopathological analysis is the standard screening modality for both ESCC and EAC. However, this is an invasive method that fails to yield a molecular profile of the diseased compartment. To decrease the invasiveness of the procedures for diagnosis, researchers are proposing non-invasive biomarkers for early diagnosis and point-of-care screening options. Liquid biopsy involves the collection of body fluids (blood, urine, and saliva) non-invasively or with minimal invasiveness. In this review, we have critically discussed various biomarkers and specimen retrieval techniques for ESCC and EAC.

## 1. Introduction

Its poor prognosis and high mortality rate make esophageal cancer (EC) one of the deadliest cancers worldwide. EC is the eighth most common cancer worldwide (the seventh most common cancer in men and the thirteenth most common cancer in women) and the sixth leading cause of cancer-related deaths. The two most common histological types of EC, namely esophageal squamous cell carcinoma (ESCC) and esophageal adenocarcinoma (EAC), account for more than 90% of ECs. The five-year survival rate of EC is less than 20% [1,2,3,4]. A recent retrospective study reporting 23,804 EAC cases and 13,919 ESCC cases suggests an increasing incidence of EAC and a decreasing incidence of ESCC in the United States [3]. ESCC is characterized by the conversion of the normal squamous esophageal epithelium to ESCC via basal cell hyperplasia, dysplasia, and invasiveness. ESCC may involve any part of the esophagus (20% upper, 50% middle, and 30% lower esophagus). Alcohol and tobacco consumption, the most common risk factors, cause cellular DNA damage and contribute to ESCC. EAC occurring in the distal esophagus occurs due to a cascade progressing from gastroesophageal reflux disease (GERD) to Barrett’s esophagus (BE), followed by EAC with columnar metaplasia playing a critical role in the pathogenesis (Figure 1). Male sex, white race, central obesity, alcohol, and smoking are common risk factors, while inflammation, genetic mutations, epigenetics, and altered microbiota play a critical role in the pathogenesis of EAC [4,5,6,7,8].

Early detection of ESCC and EAC is needed to improve survival and attenuate morbidity and mortality. Esophageal endoscopy with biopsy and histological analysis is the gold standard for early detection and diagnosis. Chromoendoscopy, virtual chromoendoscopy, magnification endoscopy, and other advanced endoscopic imaging techniques may improve the sensitivity of the detection of early-stage carcinoma [9]. However, difficulty in defining a well-characterized screening population, the lack of an accurate, cost-effective, and widely accepted screening tool, and the absence of data on the costs of non-invasive screening are associated challenges [10]. Additionally, endoscopic screening is not practical for mass screening because of the invasive and expensive procedures involved. Surgical resection is the most common treatment for early-stage EC, and chemotherapy, radiotherapy, chemoradiation, laser therapy, electrocoagulation, immunotherapy, and targeted therapy are treatment strategies for advanced and nonresectable lesions [11]. The current treatment regimen for EAC is based on the expression of a number of biomarkers, including human epidermal growth factor receptor 2 (HER2) amplification, mismatch repair deficiency/microsatellite instability (dMMR/MSI-H), and programmed death ligand 1 (PD-L1) [12]. Early-stage disease and complete resection of the lesion are favorable prognostic markers [13]. However, the ineffectiveness of chemotherapy, radiation therapy, immunotherapy, and targeted therapy contributing to low survival rates warrants the establishment of early-stage diagnostic biomarkers and novel therapeutics to improve clinical outcomes and decrease morbidity and mortality [14,15]. This notion is further supported by the asymptomatic nature of EC in its early stages, its extremely aggressive nature, and its poor survival rate. This review critically discusses the biomarkers for the early detection of EAC and the molecular aspects.

## 2. Biomarkers for EC: Pros and Cons

Poor prognosis due to late detection of ECs warrants the development of early detection methods using non-invasive biomarkers so that timely intervention can be started to improve outcomes. Tissue histology after endoscopy has limitations for mass screening, and serum tumor markers including squamous cell carcinoma antigen (SCCA) and carcinoembryonic antigen (CEA) are insufficiently specific and sensitive for early EC diagnosis [16,17]. Lesion recognition during endoscopy is impeded by inter- and intra-observer variability [18]. Blood biomarkers/liquid biopsy (circulating tumor cells, nucleic acids, tumor DNA, the tumor-derived fraction of cell-free DNA, cell-free RNA, etc.) has higher specificity and accuracy. Proteomic profiling has potential, but is limited due to higher costs in routine use, and epigenetic markers are promising due to ease of detection in tissue and body fluids including blood, plasma, and urine. Liquid biopsy is advantageous in the case of metastatic tumors which are difficult sample using a core biopsy [19]. RNA biomarkers (including mRNA, miRNA, and long non-coding RNA), protein biomarkers, metabolic biomarkers, immune biomarkers, and microbiome biomarkers are commonly documented biomarkers for EC diagnosis [20,21,22]. In addition to tissue-based or liquid biopsy-based biomarkers, imaging-based biomarkers including perfusion analysis using computed tomography (CT) or magnetic resonance imaging (MRI), texture analysis, diffusion-weighted imaging (DWI), and positron emission tomography (PET) may also be used in the treatment of EC using radiomics, an emerging field in which imaging data are converted into a high dimensional mineable feature [23]. Imaging biomarkers may have the potential to predict treatment outcomes or prognosis in EC due to their non- or less-invasive nature and wider availability.

## 3. Non-Invasive Biomarkers: Blood, Plasma, Saliva, and Urine Biomarkers

Liquid biopsy and blood biomarkers are gaining attention because of their non-invasive nature, simplicity, short-term repeatability, and cost-effectiveness, as well as their ability to detect circulating tumor cells (CTCs), circulating tumor DNA (ctDNA), and exosome-based biomarkers for both EAC and ESCC [24]. Urinary protein may serve as a biomarker for ESCC. Li et al. [25] conducted a proteomics analysis on 499 human urine samples (321 healthy individuals, 83 with ESCC, 17 with bladder cancer, 12 with breast cancer, 16 with colorectal cancer, 33 with lung cancer, and with 17 thyroid cancer). The results suggested that urinary biomarkers ANXA1, S100A8, and TMEM256 can classify ESCC, and a panel of proteins consisting of ANXA1, S100A8, SOD3, and TMEM256 is diagnostic for stage I ESCC. Further, serum expression of other factors involved in the pathogenesis of EAC and ESCC may also be potential biomarkers, e.g., chemokines and chemokine receptors. CXCL12 and its receptors CXCR4 and CXCR7 correlate with poor prognosis, CXCL10, CCL4, and CCL5 expression show anti-tumor effect, CCL20 expression is correlated with regulatory T cell recruitment, and CCR7 expression correlates with poor prognosis [26]. Additionally, neutrophil-lymphocyte ratio (NLR) [27], erythrocyte mutant frequency (EMF) [28], and serum antibodies (anti-p53, anti-p16, anti-cyclin B1, anti-c-Myc, anti-HSP70, and anti-LY6K) may potentially play a role [18] in the diagnosis of EAC and its differentiation from BE. Using saliva as a non-invasive sample for biomarkers is useful not only for oral cancers but also for non-oral cancers [29]. Liquid biopsy and blood biomarkers offer an inexpensive and non-invasive screening strategy and the use of advanced technologies, such as metabolomics and proteomics in combination, has allowed the delineation of novel diagnostic biomarkers for the early detection of ESCC and EAC [18]. However, using a single serum marker for early detection and diagnosis may have low diagnostic value, and using a panel of biomarkers in combination can significantly improve the sensitivity and specificity of the early detection and diagnosis of ESCC and EAC.

Taken together, the detection of a panel of non-invasive biomarkers in blood, urine, and saliva may increase diagnostic sensitivity and specificity and have a potential clinical application in improving outcomes [30]. Using non-invasive biomarkers in clinics will be useful because analysis of the non-invasive biomarkers utilizes readily available clinical and laboratory information to non-invasively detect the tumor early in course of disease in at-risk populations and can be applied to mass screening. Other advantages of using non-invasive strategies are the absence of adverse effects and the attenuated risk of sampling error. This will bring objectiveness to the interpretation and can overcome the limitations of endoscopy for mass screening. Further, non-invasive biomarkers are not only useful in early diagnosis, but they also play a role in predicting the treatment outcome, disease progression, and relapse [31,32]. Although the non-invasive biomarkers from saliva and urine that can be used in clinics for early detection of ESCC and EAC are limited, the results from various studies, outlined in Table 1, suggest that in addition to liquid biopsy, non-invasive samples such as urine and saliva may be used for detecting biomarkers in both ESCC and EAC.

## 4. Molecular Biomarkers

microRNA (miR), genetically conserved small noncoding RNA of 18–25 nucleotides, regulates gene expression by binding to the 3′-UTR of target mRNAs, post-transcriptionally resulting in either translational inhibition or degradation of RNA. This will cause the gene expression to be either upregulated or downregulated; activation of miRs downregulates gene expression while decreased miR expression upregulates gene expression [41,42]. The involvement of miRs in EC tumorigenesis and progression and their identification as a biomarker in blood, plasma, and urine suggests that miRs may be a potential non-invasive biomarker [41]. Fassan et al. [43] reported upregulation of miR-92a-3p, miR-151a-5p, miR-362-3p, miR-345-3p, miR-619-3p, miR-1260b, and miR-1276 as well as downregulation of miR-381-3p, miR-502-3p, and miR-3615 in the serum of early EAC patients compared with non-dysplastic BE. Further, Chiam et al. [44] reported that the ratios of RNU6-1/miR-16-5p, miR-25-3p/miR-320a, let-7e-5p/miR-15b-5p, miR-30a-5p/miR-324-5p, and miR-17-5p/miR-194-5p in circulating exosomes with an AUC of 0.99 could differentiate between EAC and nondysplastic BE. When looking for small extracellular vesicle microRNAs as biomarkers for EAC, the serum is more suitable than the plasma [45]. Thus, miRs are useful as biomarkers for diagnosis and, before surgery, to predict chemotherapy outcomes [46].

Circulating tumor DNA (ctDNA), the DNA coming out from cancerous cells and tumors and circulating in the blood, may be a potential biomarker for early diagnosis. However, detection of ctDNA in the early stages of EAC is challenging and may have limited diagnostic application [47]. Further, ctDNA levels and the detection of its variants were also found to be associated with poor survival, and the variant frequency increased with recurrence [48]. The potential of ctDNA as a biomarker and its use to monitor improvement and relapse was supported by the detection of a suitable number of somatic single-nucleotide variants (SNVs) and copy number alterations (CNAs) in the plasma of EAC patients using sequencing and a NanoString Counter [49]. Further, the detection of post-operative ctDNA provides a molecular window before the onset of overt disease and allows us to add another therapy to improve outcomes. These studies suggest the potential clinical utility of ctDNA as a prognostic biomarker for early diagnosis, monitoring treatment response and disease recurrence, and improving survival with moderate sensitivity and high specificity. This ability is further enhanced when combined with current imaging methods [50]. Additionally, cell-free plasma DNA and exosome-associated DNA from blood [19] may also be used as a biomarker for EAC diagnosis. Circulating cell-free DNA has diagnostic value and targets tumor-specific genomes by detecting epigenetic (methylation of *APC, CDKN2A, TAC1*, and *MSH2*) and genetic alterations that might have translational and clinical significance and may be more reliable than the existing biomarkers such as CEA [51]. Additionally, circular RNAs (circRNAs), which play a role in cell proliferation, migration, death, tumor invasion, and metastasis, may also be used as biomarkers for ESCC because dysregulated expression of circRNA is associated with the pathogenesis of ESCC and it can be detected not only in tumor tissue but also in nearby tissue. The detection of circRNAs using techniques such as RNA sequencing and bioinformatics analysis enables the detection of both known and unknown circRNAs, which is beneficial compared with the microarray technique, which detects only known circRNA [52]. In addition to circRNA, miRNAs, and ctDNAs, transcription factors (TFs), the regulators of gene expression, may also serve as biomarkers for early detection. TFs, including BRCA1, SOX10, ARID3A, ZNF354C, and NFIC, play a role in carcinogenesis and the development of ESCC, while SREBF1 and TFAP2A correlated with longer overall survival in ESCC. These TFs may also serve as diagnostic biomarkers [53]. Various studies [30,54] and reports, summarized in Table 2, indicate the role of microRNAs, tRNA-derived small RNAs, circulating tumor (ct) DNAs, and transcription factors as biomarkers in esophageal carcinoma.

The studies discussed in Section 3 and Section 4 suggest that the non-invasive biomarkers have a promising future in the early detection of EAC and ESCC. However, which strategy should be used in the long term is a topic for discussion. A meta-analysis study by Wong et al. [63] including 161 studies and 32,209 subjects concluded that cytology and endoscopy had a comparable diagnostic accuracy when detecting autoantibody and microRNAs as future diagnostic biomarkers. Further, just as a single protein or gene has low diagnostic value and a panel of biomarkers can increase the predictive value, a panel of microRNAs and isoforms of microRNAs formed using next-generation sequencing (NSG) may serve as novel biomarkers with increased sensitivity.

## 5. Imaging-Based Biomarker

The accurate staging of the EC is important when determining the most suitable treatment strategy, and radiological assessment serves the purpose. TNM staging is based on magnetic resonance imaging (MRI), multidetector computed tomography (MDCT), and endoscopic ultrasonography (EUS). However, CT-positron emission tomography (PET) has a limited role, but MRI-PET has better accuracy [64]. The diagnostic sensitivity of esophageal endoscopy and biopsy for the early detection of ESCC or EAC can be increased by novel endoscopic techniques, including dye spray chromoendoscopy, virtual chromoendoscopy (VCE), confocal laser endomicroscopy (CLE), and volumetric laser endomicroscopy (VLE) [18,65]. Additionally, deep learning based on analyzing images and videos has great potential to accurately and quickly diagnose esophageal cancer [66]. The studies on the use of imaging as a diagnostic modality are limited, and most of the studies have reported their prognostic value. Mizumachi et al. [67], based on a study involving 109 ESCC patients, reported that intravoxel incoherent motion magnetic resonance imaging (IVIM-MRI), which can quantify micro-perfusion at the capillary level in the tissue, may be used as a non-invasive prognostic biomarker for ESCC, reflecting clinical stage and survival. Wen et al. [68], using data from 220 ESCC patients in a retrospective study, assessed the potential value of classifying patients according to PD-L1 and CD8+TIL expression levels with CT-based radiomics and suggested that predictive performance can be significantly increased by a combination of clinical factors and radiomics signatures. Zeng et al. [69] reported the potential of contrast-enhanced computed tomography (CT)-based imaging biomarkers (IBMs) in predicting treatment outcomes in terms of overall survival and progression-free survival in ESCC patients after chemoradiotherapy. Further, the prognostic value of diffusion-weighted magnetic resonance imaging (DWI) in predicting the survival of RSCC patients [70] and the potential of ^1^H nuclear magnetic resonance spectroscopy (^1^H-NMR) in detecting early-stage ESCC and differentiating pre- and post-operative ESCC, evaluating the therapeutic response of surgery, and monitoring postoperative chemoradiotherapy responses [71] suggest the potential of imaging strategies for staging and prognosis. A recently developed strategy involving CT and MRI radio-omics has been shown to be useful in determining metastasis [64]. However, most of the studies reported in the literature have documented the role of imaging modality in staging, prognosis, and evaluating treatment response or relapse, but there are limited studies delineating its utility for the early detection of ESCC or EAC, and therefore this warrants further research.

## 6. Druggable Targets

The incidence rate of esophageal adenocarcinoma (EAC) is one of the fastest rising in the US, yet the prognosis is poor, as many patients present to their oncologists at the advanced stages of disease progression [72]. One of the reasons for this may be the subpar standard of care for EAC monitoring. Currently, patients undergo endoscopy, where a tissue sample from a pinch biopsy is stained for antibody detection and undergoes gross inspection with a microscope. However, it has been illustrated that the current method is not precise enough to identify patients at risk for EAC. This is occurring because pre-cancerous tissues of disparate morphologies are at times labeled as having the same level of expression of cancerous markers. Barrett’s esophagus (BE) is a precursor to EAC, and gastroesophageal reflux disease (GERD) is a precursor to BE (Figure 1). Therefore, early detection of EAC may be made possible by identifying individuals with GERD or BE with high expression of oncogenic markers found in either Barrett’s tissues or esophageal carcinomas.

### 6.1. Human Epidermal Growth Factor Receptor (HER)2

Studying the presence and levels of biomarkers in BE tissue that progress to EAC is one way clinical scientists have been testing the utility of their diagnostic panels of markers. A collaborative study presented at ASCO-GI in 2022 by the Mayo Clinic and Kansas University Medical Center demonstrated insignificant expression levels of HER-2 and PD-L1 in 50+ non-progressive and progressive BE tissues as well as esophageal tumor tissues assessed with mass spectrometry [73]. However, the only FDA-approved EAC therapeutics use these exact biomarkers—HER2 and PD-L1—as their targets. The discrepancy in the biomarker targets of the current EAC therapy and the lack of significant expression in BE patients who are eligible to undergo these therapies illustrates the need to develop more accurate targets for EAC diagnostics and treatment regimens.

Trastuzumab, commercially known as Herceptin, is a Roche-developed HER2-targeted therapy that can directly treat HER2+ cancer cells, which are commonly overexpressed in breast and stomach cancer and have also been suggested to be present in esophageal cancer [74]. In EAC specifically, however, the literature has suggested that the presence of HER2+ in EAC ranges between 15% and 29% [75]. While this humanized monoclonal antibody is generally well tolerated by patients with HER2+ cancers, trials revealed that the therapy in combination with chemotherapy led to an increased incidence of cardiotoxicity [76]. Due to the relatively low presence of HER2+ in EAC and tucotuzumab’s potential for cardiotoxicity, it is evident that patients with EAC need a companion diagnostic test before beginning this therapy. It should be noted, however, that Roche adopted biomarker testing guidelines for breast cancer patients, though for no other Herceptin-prescribed cancer types. This companion diagnostic was approved by the FDA in July 2020.

### 6.2. Programmed Death Ligand (PD-L)1

In the past decade, the understanding of the mechanisms by which tumors evade the immune system has led to significant breakthroughs in cancer treatment options. The development of immune checkpoint inhibitor (ICI) therapies is the most notable example. A key mechanism pathway leading to tumor immune evasion is the interaction of the well-known immune checkpoints CD279 (programmed cell death protein 1, (PD-1), expressed on immune cells) and CD274 (programmed death ligand 1, (PD-L1), expressed on tumor cells). Clinical trials have shown the effectiveness of anti-PD-1 and anti-PD-L1 antibodies in the treatment of many cancerous indications, including esophageal cancers. Currently, PD-L1 immunohistochemistry (IHC) and tumor mutational burden (TMB) are approved by the FDA as companion diagnostics for immunotherapy, although neither modality has shown a correlation to outcome in all cases [77,78,79].

Immune checkpoint inhibitor (ICI) therapies have disadvantages, including low and varied response rates (ORR for most indications is ≤30%), risks of immune-related adverse events (17%, Grade 3 or higher), and an observed acquired resistance [80]. ICIs are one of the most expensive classes of anti-cancer therapies, and it has been well-described in the literature that the efficacy of this modality is very low, irrespective of tissue-based companion diagnostics which attempt to stratify responses to therapies used to initiate programmed cell death. There is an enormous healthcare and economic benefit opportunity in stratifying more accurately those who will have a robust response to immunotherapy versus those who will not achieve a therapeutic benefit (regression of primary tumor and/or metastatic deposits).

PD-L1 histopathology presents technical and clinical challenges when determining which patients should be placed on ICIs. The testing sample must come from a slice of primary tumor tissue that has been unadulterated by chemoradiation. Interpretation is strongly dependent on cancer type, treatment indication, IHC scoring method, and pathologist experience level. Technical performance is strongly dependent on the antibody clone (five clones for five ICIs), staining platform, fixation time, and fixation type. The method is dependent on antibody binding and epitope integrity, proper tissue fixation is subject to inter- and intra-observer disagreement, and discordant results across clinical trials have been reported. Additionally, defining a high TMB is not currently possible across various cancers.

Pembrolizumab, commercially known as Keytruda, is a Merk-developed anti-PD-1 treatment intended for patients with esophageal or gastroesophageal junction (GEJ) carcinoma. Keytruda was approved by the FDA on March 23, 2021 in response to promising results from the KEYNOTE-590 clinical trial [81,82]. However, targeting PD-L1 for EAC therapy may not be efficient, as fewer than half of the participants (42%; 109 out of 259 patients) were PD-L1 negative, and 93.6% of these participants did not have a significant response to pembrolizumab [83]. Despite FDA approval of the drug, assessments of pembrolizumab have demonstrated that patients with ESCC have a higher combined positivity score (CPS) than EAC patients.

Additionally, in an early clinical trial, KEYNOTE-181, Keytruda reported that high-CPS-scoring EAC patients had improved overall response rates (ORRs) to pembrolizumab. However, the duration response results for said participants did not improve compared with those with ESCC [84]. It is presumed that the success of this trial was ramped up by ESCC and PD-L1 CPS > 10 tumors [85]. Notably, previous assessments of PD-L1 in EAC tumors have led to much scrutiny, with researchers claiming PD-L1 presence in only 27% of EAC tumors, or even lower [86]. The variation between ESCC and EAC PD-L1 positivity may therefore be a large contributing factor to the low response rate for pembrolizumab in EAC patients.

### 6.3. Epidermal Growth Factor Receptor (EGFR)

Cetuximab (CET), commercially known as Erbitux, is an Eli Lilly and Company-owned chemotherapeutic that works as an epidermal growth factor receptor (EGFR) inhibitor. While the monoclonal antibody has been shown to improve the outcomes of head and neck cancers, a meta-analysis performed in 2019 “did not reveal that CET could significantly contribute to the increase of overall survival and PFS (1–5 years) in localized esophageal carcinoma” [87]. Despite these results, the literature has indicated that EGFR is overexpressed in approximately 20–50% of EAC tumors. Ultimately, the meta-analysis performed by Ze-Hao Huang et al. suggested “that adding CET to multimodal therapy significantly improved response rate and disease control rate for patients with metastatic esophageal cancer instead of patients with localized esophageal cancer. CET might be a safe therapeutic choice, but CET failed to significantly improve the overall survival and PFS for patients with localized or metastatic esophageal cancer” [87].

### 6.4. Programmed Death Ligand (PD-L)2

In addition to PD-L1, PD-L2 (another ligand to the PD-1 receptor which regulates cell division) has been highly scrutinized in clinical studies of esophageal cancer therapies. Some studies have assessed the presence of significant PD-L2 expression in esophageal cancer [88,89,90]. One study in particular claims to have identified epithelial PD-L2 expression in 51.7% of esophageal adenocarcinomas [88]. It is important to note that the reported result of 51.7% represents samples with an IHC score of 1 or more [91]. A study by Derks et al. used IHC techniques and a scoring system like that of Herceptin marker expression technology [88]. A 2020 study found that the majority (83.8%) of participants were PD-L1 negative, and another previous clinical study observed that PD-L2 expression has an inverse relationship with the presence of cancerous T-cells [89,90]. These results suggest a need for the careful evaluation of current PD-L2-targeting esophageal cancer therapies.

### 6.5. New Exploratory Markers

EAC has been found to resist many first-line chemotherapy tactics, including but not limited to cisplatin, 5-FU, taxol, and carboplatin. In addition, one study has suggested that there is indeed no survival advantage for EAC patients treated with surgery and chemotherapy versus surgery alone [92]. Mittal, et al. and Hartley, et al. presented research throughout 2022 to shed light on the proteomic environment that is contributing to the rapidly metastatic and chemotherapy-resistant cancer that is esophageal adenocarcinoma [73]. With the ever-growing public health concerns, this area of research will be important in establishing targets and diagnostic tools such that we can better fight such poor prognoses. Unadulterated EAC samples from esophagectomies of patients who had never received them were processed and microdissected, digested, and analyzed with a mass spectrometry platform (Triple-quad). Their data produced highly significant and interesting targets related to chemotherapy resistance and sensitivity, apoptosis suppression, and proliferation upregulation. Four main drivers (DAD1, ISG15, S100P, and UBE2N) were deemed to be of potential immediate and high impact because of their many intersections with diagnostic and therapeutic aspects of molecular oncology (Figure 2). Notably, overexpressed DAD1 protein levels contribute to cisplatin resistance, which is striking since 95% of EC patients treated after esophagectomy were placed on cisplatin by their community oncologist in the first-line setting [93]. S100P expression has been found to increase resistance to 5-FU in colorectal cancer [94], and 5-FU was prescribed to >70% of EC patients, so if there is innate resistance to the two most prescribed drugs for esophageal cancer then alternative patient management strategies are needed [95].

DAD1, ISG15, S100P, and UBE2N have important microbiological functions in human physiology. S100 proteins are involved in cellular regulatory processes, managing proliferation, differentiation, and apoptosis, usually in response to inflammation or cell damage. Ubiquitins are also regulatory proteins called into action to mark damaged cells for destruction and signal for cellular proliferative measures to patch up areas of injured tissue, typically during inflammation or after bile-acidic assaults [96,97,98,99,100,101,102,103,104,105,106,107,108,109,110,111]. These mechanisms are the body’s natural response to areas of the body in need of cellular debris removal and reconstruction. When these processes are dysfunctional or overactive, carcinogenesis can occur. Currently, the BE-smart assay (ProPhase Laboratories, Garden City, NY, USA) is a mass spectrometry-based test that offers a fully quantitative analysis of these markers and other proteins indicative of disease progression from formalin-fixed, paraffin-embedded forceps biopsies [112]. These newly emerging biomarkers may demonstrate the predictability of carcinogenesis, as traditional oncogenes have failed to yield consistent prognostic results.

## 7. Artificial Intelligence

Artificial intelligence (AI), the simulation of human intelligence processes by machines, comprises perceiving, synthesizing, and inferring information, and giving the output. AI has been used to improve the diagnosis of various upper gastrointestinal diseases, and meta-analyses have shown promising results for the diagnosis of ESCC and EAC with high sensitivity and specificity [18]. Due to the lack of tissue specificity of the gene biomarkers and the scarcity of biomarkers, deep learning processing imaging, videos, and convolutional neural networks interpreting endoscopic images and videos may be capable of excellent performance in evaluating early esophageal neoplasia with sensitivity and specificity [113]. AI may be useful in the mass screening of the population while avoiding unnecessarily invasive procedures in the early detection of the neoplastic lesion and management of Barrett’s esophagus. However, a further development and spread of AI are needed for its routine use and to decrease morbidity and mortality in EC [114]. The meta-analyses performed using AI were mostly retrospective studies using endoscopic images, though some recent studies have carried out analyses using real-time images and videos [18,115,116,117] and have shown the potential of AI for predicting submucosal invasion and differentiating stage T1a from T1b with an accuracy, sensitivity, and specificity of 71%, 77%, and 64%, respectively [118]. Further, using 48 clinically proven molecules associated with ESCC progression and machine learning, Li et al. [119] identified stratifin (SFN) as an optimal prognostic biomarker for ESCC. Similarly, machine learning applied to GSE20347, GSE38129, GSE75241, and TGCA datasets has identified the diagnostic biomarkers GPX3, MMP1, and MMP12 associated with immune cell (CD8+ T cells, M0 and M2 macrophages, and dendritic cells) infiltration in ESCC [120]. These results and other studies suggest that AI may be used to identify diagnostic biomarkers, predict five-year survival in EC [121], and facilitate surveillance of Barrett’s esophagus [122], although further research is warranted.

## 8. Bioinformatics/In Silico Analysis

Bioinformatics is another tool to identify potential diagnostic and prognostic biomarkers for EC (EAC and ESCC). An analysis of differentially expressed genes (DEGs) between BE and EAC using two public databases (GSE26886 and GSE37200) revealed 27 upregulated genes and 104 downregulated genes, and these genes were found to be involved in tumorigenesis in gene ontology (GO) and gene set enrichment analysis (GSEA). Among 5 upregulated genes (*MYO1A*, *ACE2*, *COL1A1*, *LGALS4*, and *ADRA2A*) and 3 downregulated genes (*AADAC*, *RAB27A*, and *P2RY14*), expression of *AADAC*, *ACE2*, and *ADRA2A* showed significant correlation with patients’ survival probability [123]. Another study compared the gene profile of 52 EACs (tubular EAC), 70 gastroesophageal junction adenocarcinomas (GEJACs), 8 normal esophageal mucosa, and 5 normal gastric mucosa samples using principal component analysis, hierarchical clustering, and survival-based analyses and revealed extensive similarity and related molecular processes between tubular EAC and GEJAC. These results suggest that identical cell surface markers/genes (e.g., *CDH11*, *ICAM1*, and *CLDN3*) overexpressed in both Barrett’s-derived EAC and without Barrett’s metaplasia may be used to simultaneously detect both subtypes [124]. Another study using the TGCA database performed differential gene analysis, survival statistical analysis, miRNA–mRNA, and protein interaction network analysis and reported that five miRNAs (miR-18a, miR-29c, miR-181b, miR-345, and miR-615) and seven genes (*AASS*, *AKAP6*, *ARHGAP24*, *ESM1*, *FABP3*, *GK*, and *NFIX*) were statistically correlated with the survival of patients with EC [125]. Further, Zhang et al. [53], using a total of ten published microarray ESCC datasets, assessed the role of the transcription factors (TFs) BRCA1, SOX10, ARID3A, ZNF354C, NFIC, SREBF1, and TFAP2A in ESCC, which can be used as biomarkers for early detection. Further, the detection of the transcription factors ELF3, KLF5, GATA6, and EHF promoting each other’s expression by interacting with each super-enhancer in EAC tissues [126] supports the notion that TFs may be used as biomarkers.

## 9. Biomarkers: Correlation with Survival

Human papillomavirus (HPV) infection is related to Barrett’s dysplasia and EAC, and, thus, the association of HPV with other biomarkers may have prognostic significance. Rajendra et al. investigated the survival rates of 142 patients with or without HPV infection and found that Barrett high-grade dysplasia and EAC patients positive for HPV had a favorable prognosis compared with those who were HPV negative [127]. In a study involving 143 patients evaluating the prognostic significance of HPV-related biomarkers, including retinoblastoma protein (pRb), cyclin D1 (CD1), minichromosome maintenance protein (MCM) 2, and Ki-67, in Barrett high-grade dysplasia and EAC reported that only low levels of cyclin D1 had an association with a favorable prognosis for overall survival. The study concluded that CD1 is an independent prognostic marker in Barrett high-grade dysplasia and EAC, and that HPV positivity in combination with pRb, CD1, MCM2, and Ki-67 correlate with a survival benefit in EC [128].

## 10. Conclusions

Although tissue biopsy during esophageal endoscopy followed by histopathological analysis is the standard diagnostic method for ESCC and EAC, the need for non-invasive early markers for mass screening is obvious. This is because tissue biopsy during endoscopy is inappropriate for use in mass screening. Recent research, as discussed above, suggests that the use of liquid biopsy, urine, and saliva may serve as non-invasive methods for investigating biomarkers for ESCC and EAC. The advantage will be that these samples can be used for mass screening. Further, longitudinal serial monitoring of the liquid biopsy is important in treatment response, relapse prediction, and prognosis for CT responses in advanced ESCC [129]. Additionally, artificial intelligence and the in silico analysis of existing data will aid in developing more definitive early biomarkers for ESCC and EAC, though much research is needed to apply AI and in silico analyses in clinics.

## Figures and Tables

**Figure 1 ijms-24-03316-f001:**
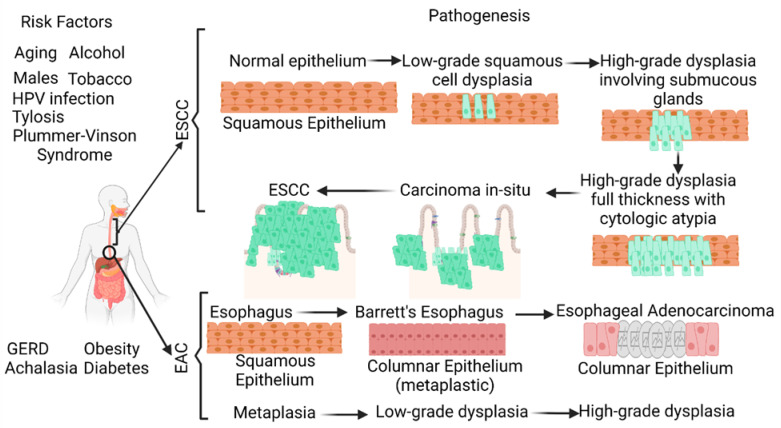
Schematic diagram showing the risk factors and pathogenesis of esophageal adenocarcinoma (EAC) and esophageal squamous cell carcinoma (ESCC). ESCC develops through a multistep process from basal hyperplasia due to chronic esophagitis through increasing severity of dysplasia, while EAC develops through a cellular cascade involving gastroesophageal reflux disease (GERD) followed by Barrett’s esophagus leading to EAC. There are different risk factors in the pathogenesis of ESCC and EAC. HPV, human papillomavirus.

**Figure 2 ijms-24-03316-f002:**
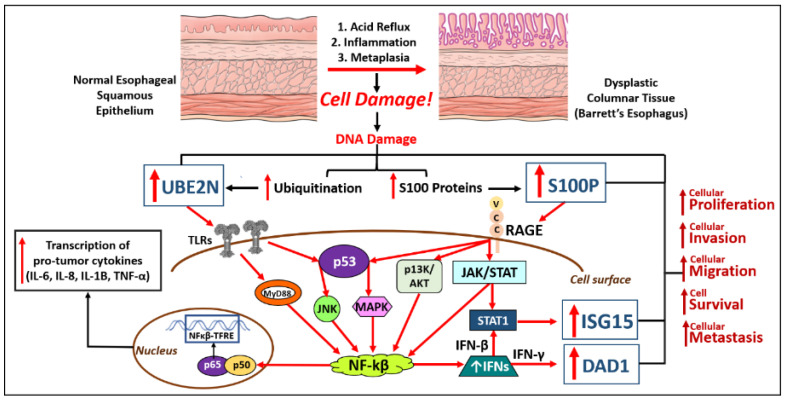
Biochemical malignant transformation of distal esophageal compartment during the chronic acidic assault. UBE2N and S100P are upregulated during the body’s well-intentioned response to repair cell damage in the distal esophagus mucosa, leading to unfavorable pro-tumoral processes. It should be noted that these proposed interconnecting pathways have been assembled from over a dozen references and should be considered mostly hypothetical, with the hope of future experiments to further elucidate and confirm the mutual intersections and oncogenic mechanisms of these four markers during esophageal adenocarcinoma pathogenesis. UBE2N (ubiquitin-conjugating enzyme E2 N), RAGE (receptor for advanced glycation end products), IL (interleukin), TNF-α (tumor necrosis factor-alpha), NF-κB (nuclear factor kappa beta), MyD88 (myeloid differentiation primary response 88), JNK (c-Jun N-terminal kinases), MAPK (mitogen-activated protein kinases), PI3K (phosphoinositide 3-kinases), AKT (protein kinase B), IFN-γ (interferon gamma), JAK (Janus kinase), STAT (signal transducer and activator of transcription), ISG15 (interferon-stimulated gene 15), DAD1 (defender against cell death 1).

**Table 1 ijms-24-03316-t001:** Non-invasive biomarkers for esophageal carcinoma. Esophageal squamous cell carcinoma (ESCC), esophageal adenocarcinoma (EAC), gelsolin (GSN), serum paraoxonase/arylesterase 1 (PON1) and serum paraoxonase/lactonase 3 (PON3), desmoglein-2 (DSG2), serum amyloid A1 (SAA1), enolase 1 (ENO1), triosephosphate isomerase 1 (TPI1), toll-like receptor (TLR)-4, hypoxia-inducible factor (HIF)-1α, tricarboxylic acid (TCA) cycle, deoxynivalenol (DON), neosolaniol (NEO), T-2 toxin (T-2), HT-2 toxin (HT-2).

Sample Type	EAC/ESCC	Sample Size	Biomarker Type/Observation
Serum [33]	EAC	159 EAC patients	Metabolomic profiling; among D-mannose, L-proline (LP), and 3-hydroxybutyrate (BHBA) were significantly different in the EAC patients and in the controls; the serum level of D-mannose may be a novel prognostic biomarker for EAC
Serum [34]	EAC	301 samples	To identify glycoprotein biomarkers; different glycoforms of complement C9 (C9), GSN, PON1, and PON3 are biomarkers for EAC and discriminate it from BE; serum levels of C9 glycoforms increase with disease progression
Saliva [35]	EAC	DNA methylation profiles for 125 EAC and 64 normal adjacent squamous samples; saliva samples from 192 patients	A proto-cadherin module centered around CTNND2 is inactivated in Barrett’s esophagus; CCL20 chemokine methylation pattern in saliva correlates with EAC status
Serum [36]	ESCC, EJA	151 ESCC and 96 EJA cases with 212 healthy controls.	Serum DSG2 was significantly higher in ESCC and EJA compared with controls; serum DSG2 levels were significantly associated with patient age and histological grade in ESCC; serum DSG2 may be a biomarker for ESCC and EJA
Serum [37]	ESCC	30 ESCC patients and 30 healthy controls	Serum proteins S100A8/A9, SAA1, ENO1, TPI1, and PGAM1 have high diagnostic sensitivity and specificity for ESCC; glycolysis, TLR4, HIF-1α, Cori cycle, TCA cycle, folate metabolism, and platelet degranulation are commonly deregulated pathways
Saliva [38]	ESCC	178 ESCC patients and 101 healthy controls	Significantly higher numbers of *Streptococcus salivarius*, *Fusobacterium nucleatum,* and *Porphyromonas gingivalis* in patients with ESCC suggest salivary microbiota as a biomarker
Urine [25]	ESCC	499 urine samples (83 ESCC)	ANXA1, S100A8, and TMEM256 can classify ESCC; a combination panel of the proteins ANXA1, S100A8, SOD3, and TMEM256 is diagnostic for stage I ESCC
Serum [39]	EC	20 EC patients and 20 healthy controls	Serum anaphylatoxin C3a may be a promising biomarker in the diagnosis of EC
Urine [40]	EC	10 controls and 17 EC patients	Mycotoxins as binary (NEO/HT-2 and T-2/HT-2) and ternary (DON/NEO/HT-2) combinations were present in the urine samples of patients with EC

**Table 2 ijms-24-03316-t002:** Molecular biomarkers for ESCC and EAC. Area under the curve (AUC), esophageal adenocarcinoma (EAC), esophageal squamous cell carcinoma (ESCC), healthy control (HC), circulating tumor (ct) DNA, Barrett’s esophagus (BE), next-generation sequencing (NGS), variants of unknown significance (VUS), gelsolin (GSN), serum paraoxonase/arylesterase 1 (PON1) and serum paraoxonase/lactonase 3 (PON3), esophageal squamous cell carcinoma (ESCC) and esophagogastric junction adenocarcinoma (EJA), desmoglein-2 (DSG2), tRNA-derived small RNAs (tsRNAs).

Sample Type	EAC/ESCC	Sample Size	Biomarker Type/Observation
Urine [41]	EAC and ESCC	150 HCs and 43 ESCCs 144 HCs and 8 EACs	Significantly higher miR-1273f, miR-619-5p, miR-150-3p, miR-4327, and miR-3135b levels in ESCC and EAC compared with HCs; miR-1273f and miR-619-5p with AUC ≥ 0.80 for diagnosing stage I ESCC, AUC ≥ 0.80 in ESCC, and AUC = 0.80 for EAC
Urine, saliva, and blood [55]	ESCC	72 ESCC patients	Serum cell-free miR-1246 expression in the urine, saliva, and serum may be a useful biomarker for ESCC and urine can be used as a non-invasive sample instead of blood
Plasma [56]	ESCC	16 healthy controls and 66 ESCC patients	Plasma miR-21, miR-31, and miR-375 could be potential biomarkers for the diagnosis of ESCC, while miR-31 and miR-375 have sufficiently high sensitivity and specificity to differentiate ESCC patients from healthy controls
Saliva [57]	EC	miRNA expression profile GSE41268	miR-144, miR-451, miR-98, miR-10b, and miR-363 may serve as biomarkers for EC
Saliva [58]		32 EC patients and 16 healthy controls	Salivary supernatant miR-21 was significantly higher in EC with a sensitivity and specificity of 84.4% and 62.5%, respectively; miR-21 expression does not correlate with EC stage
Saliva [59]	EC	7 EC patients and 3 healthy controls	miR-10b*, miR-144, and miR-451 in the whole saliva and miR-10b*, miR-144, miR-21, and miR-451 in saliva supernatant were significantly upregulated in patients, with sensitivity and specificity ranging between 43.6% and 92.3%
Saliva [60]	ESCC	3 ESCC patients and 3 healthy controls	RNA sequencing of salivary exosomes for identification of tsRNA; tsRNA (tRNA-GlyGCC-5) was significantly enriched in salivary exosomes in ESCC
Plasma [47]	EAC	Patients with stage I to IV EAC 55 tumor and matched normal samples	Detection frequency and quantity of ctDNA increase with stage; ctDNA positively correlates with disease burden; ctDNA levels during the treatment may be useful to determine response and recurrence in some patient
Plasma [48]	EAC	209 blood and tumor samples from 57 EAC patients	Both plasma and tumor samples were sequenced for ctDNA; detectable ctDNA variants in post-treatment plasma samples were associated with worse disease-specific survival; variant allele frequency of ctDNA variants increased with disease recurrence
Plasma [61]	BE EAC	138 patients: EAC = 41 Barrett’s dysplasia = 48 Control = 49	To detect circulating HPV DNA; higher circulating HPV DNA was detected in EAC patients with invasive tumors with submucosal invasion and lymph node metastasis; circulating HPV DNA positivity was associated with tissue HPV positivity and disease severity
Plasma [49]	EAC	40 EAC patients (17 palliative and 23 curative)	Sensitive ctDNA detection has potential for the monitoring and predicting of short overall survival; the presence of ctDNA post-operatively predicts relapse and provides a molecular window before the onset of overt disease
Plasma [62]	EAC	55 EAC patients with advanced disease	ctDNA detection using NGS; 66% of patients had ≥ 1 genomic alteration including VUS and 69.1% had ≥ 1 characterized alteration (excluding VUSs); patients with ≥ 1 characterized alteration had alterations targetable by an FDA-approved therapy theoretically

## Data Availability

Not applicable since the information is gathered from published articles.

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
