# Peer review of "Biomarkers for Early Detection, Prognosis, and Therapeutics of Esophageal Cancers"

_ijms, 2023, doi:10.3390/ijms24043316_

Round 1

Reviewer 1 Report

Review article has detailed explanation of all the biomarkers available for EC and ESCC. This is a fine piece of work and can be published.

Author Response

Comment: Review article has detailed explanation of all the biomarkers available for EC and ESCC. This is a fine piece of work and can be published.

Response: Thank you for your comment.

Reviewer 2 Report

The review article “Biomarkers for early detection of esophageal cancer” by Rai et al. aims to provide  insight into promising potential new diagnostic markers which could be assessed non-invasively compared to current standard endoscopy screenings.

This review is timely as esophageal cancers, especially EAC patient numbers are stagnant and therapies have not provided much improvement for survival rates. A major weakness is that the introduction and Figure 1 do not sufficiently compare and contrast that ESCC and EAC are two different diseases. A substantial overall flaw is that each paragraph lists different experimental research and/or clinical studies and their outcome without providing context what this means for the current state of knowledge (no commentary is included, just a laundry list of findings) and how it is possibly applied in the patient setting, if at all. Many of the markers identified are currently assessed using endoscopically retrieved biopsies, so the claim for non-invasive screening is not sufficiently addressed here.

Major points to address:

The introduction as well as Figure 1 do not provide enough information to identify which biomarkers, for example, apply to which type of cancer. The initial description provides the characterization of ESCC vs EAC but then HER2, dMMR/MSI-H and so on are listed as if they apply to both. This should be clarified. 

More so, Figure 1, on the left is incorrect: arrows for ESCC need to by pointing to the proximal esophagus and EAC to the distal (closer to the stomach). The matching risk factors are also incorrect, as GERD and obesity are associated with EAC, but HPV, alcohol and tobacco preferentially with ESCC. This is very misleading in its current form although the text is more specific. 

The title should be reconsidered: the biomarkers discussed are presented in the context of treatments and highlight attempts for personalized or targeted medicine. The ‘new exploratory marker’ section refers again to markers/drivers of chemotherapy resistance and recurrence, not so much to biomarkers. Other research studies have identified common drivers of ESCC and EAC, as well as modifications frequent between other cancers and more unique to ESCC and EAC. These would be helpful in the evaluation, then more context of function should be provided for recently identified markers such as DAD1, ISG15, S100P, and UBE2N. Not enough context is provided for the ubiquitin proteins: are they currently targeted as therapy or are they used for diagnostics?

After that, the authors highlight other relevant markers, only that most of these are currently detected and evaluated on tissue samples. So, how would that address the advantage of the non-invasive sampling and diagnosis referred to in the introduction? AI, bioinformatics, all are dependent on tissue sampling. The only paragraph relating to non-invasive sampling gives a summary of circulating DNAs and other liquid biopsies analyzed experimentally but no discussion for its reliability and potential clinical application is provided. That should be expanded.

Minor: 

Formatting: comments from other authors are still included in review formatting.

Figure 1 has a title but no legend

Figure 2 has a legend but not a title

Page 13, line 274 no ending of the bracket for sample composition.

Author Response

Comments: The review article “Biomarkers for early detection of esophageal cancer” by Rai et al. aims to provide insight into promising potential new diagnostic markers which could be assessed non-invasively compared to current standard endoscopy screenings. This review is timely as esophageal cancers, especially EAC patient numbers are stagnant and therapies have not provided much improvement for survival rates.

Response: Thank you for your comments and suggestions.

Concern 1: A major weakness is that the introduction and Figure 1 do not sufficiently compare and contrast that ESCC and EAC are two different diseases. A substantial overall flaw is that each paragraph lists different experimental research and/or clinical studies and their outcome without providing context what this means for the current state of knowledge (no commentary is included, just a laundry list of findings) and how it is possibly applied in the patient setting, if at all. Many of the markers identified are currently assessed using endoscopically retrieved biopsies, so the claim for non-invasive screening is not sufficiently addressed here.

Response: Thank you for your comments and suggestions. We have modified the text and Figure 1. The revised manuscript differentiates the text for ESCC and EAC and also highlights the limitations of endoscopic biopsy and the need for non-invasive biomarkers.

Concern 2: The introduction as well as Figure 1 do not provide enough information to identify which biomarkers, for example, apply to which type of cancer. The initial description provides the characterization of ESCC vs EAC but then HER2, dMMR/MSI-H and so on are listed as if they apply to both. This should be clarified.  More so, Figure 1, on the left is incorrect: arrows for ESCC need to by pointing to the proximal esophagus and EAC to the distal (closer to the stomach). The matching risk factors are also incorrect, as GERD and obesity are associated with EAC, but HPV, alcohol and tobacco preferentially with ESCC. This is very misleading in its current form although the text is more specific. 

Response: Thank you for your comments and suggestions. We have modified the text and Figure 1. The revised manuscript differentiates the text for ESCC and EAC and highlights the limitations of endoscopic biopsy and the need for non-invasive biomarkers.

Concern 3: The title should be reconsidered: the biomarkers discussed are presented in the context of treatments and highlight attempts for personalized or targeted medicine. The ‘new exploratory marker’ section refers again to markers/drivers of chemotherapy resistance and recurrence, not so much to biomarkers. Other research studies have identified common drivers of ESCC and EAC, as well as modifications frequent between other cancers and more unique to ESCC and EAC. These would be helpful in the evaluation, then more context of function should be provided for recently identified markers such as DAD1, ISG15, S100P, and UBE2N. Not enough context is provided for the ubiquitin proteins: are they currently targeted as therapy or are they used for diagnostics?

Response: Thank you for your suggestion and comments. We have modified the title as suggested.

Concern 4: After that, the authors highlight other relevant markers, only that most of these are currently detected and evaluated on tissue samples. So, how would that address the advantage of the non-invasive sampling and diagnosis referred to in the introduction? AI, bioinformatics, all are dependent on tissue sampling. The only paragraph relating to non-invasive sampling gives a summary of circulating DNAs and other liquid biopsies analyzed experimentally but no discussion for its reliability and potential clinical application is provided. That should be expanded.

Response: Thank you for your comments and suggestions. We have modified the text to highlight the need for non-invasive markers and their clinical relevance.

Concern 5: Formatting: comments from other authors are still included in review formatting.

Response: Thank you for pointing out the formatting issue. We have addressed it in the revised manuscript.

Concern 6:Figure 1 has a title but no legend.

Response: Thank you for your comment. We have included the legend in Figure 1.

Concern 7: Figure 2 has a legend but not a title.

Response: Thank you for your comment. We have added the title to Figure 2.

Concern 8: Page 13, line 274 no ending of the bracket for sample composition.

Response: Thank you for your comment. We have corrected the text.

Reviewer 3 Report

Rai et al. report in this review article highlight the prospective biomarkers for ECC, which might inspire precision treatments for ECC. The following comments were raised during the review of this paper: 

I would suggest the author re-organize the manuscripts. A better outline can improve the audience's reading experience. 

- introduction, early detection benefit, and challenge.

- noninvasive based: Blood, plasma, and urine biomarkers/metabolite; current progress

- molecular-based; gene expression, miRNA, lncRNA, methylation, etc. Current progress

- Image-based (DL), CT, image, CNN-based model, etc. Current progress

- druggable target for ECC, chemo, Immunotherapy, etc

Author Response

Comment: Rai et al. report in this review article highlight the prospective biomarkers for ECC, which might inspire precision treatments for ECC. The following comments were raised during the review of this paper: 

Response: Thank you for your comments.

Concern 1: I would suggest the author re-organize the manuscripts. A better outline can improve the audience's reading experience. 

- introduction, early detection benefit, and challenge.

- noninvasive based: Blood, plasma, and urine biomarkers/metabolite; current progress

- molecular-based; gene expression, miRNA, lncRNA, methylation, etc. Current progress

- Image-based (DL), CT, image, CNN-based model, etc. Current progress

- druggable target for ECC, chemo, Immunotherapy, etc

Response: Thank you for your suggestions. We have rearranged the text as suggested by the reviewer.

Reviewer 4 Report

  it is a multi-generational review without a particular focus and without giving any diagnostic, clinical or therapeutic approaches.

Author Response

Comment: it is a multi-generational review without a particular focus and without giving any diagnostic, clinical or therapeutic approaches.

Response: Thank you for taking the time to review our manuscript and providing feedback on the flow of the paper. We have arranged the format so the biomarkers we highlight have a clearer connection to diagnostic, clinical and therapeutic utility.

Round 2

Reviewer 2 Report

The resubmission contains a much-improved version. The authors addressed all my comments and suggestions sufficiently.

Reviewer 3 Report

No new comments for the revised version. 

Reviewer 4 Report

The authors improved their manuscript to a more specific way.